# Nanosecond Pulsed Electric Field (nsPEF): Opening the Biotechnological Pandora’s Box

**DOI:** 10.3390/ijms23116158

**Published:** 2022-05-31

**Authors:** Alvaro R. Ruiz-Fernández, Leonardo Campos, Sebastian E. Gutierrez-Maldonado, Gonzalo Núñez, Felipe Villanelo, Tomas Perez-Acle

**Affiliations:** 1Computational Biology Lab, Centro Científico y Tecnológico de Excelencia Ciencia & Vida, Fundación Ciencia & Vida, Santiago 7780272, Chile; leocampos@dlab.cl (L.C.); sebastian@dlab.cl (S.E.G.-M.); gonzalo.nunez@dlab.cl (G.N.); felipe@dlab.cl (F.V.); 2Facultad de Ingeniería y Tecnología, Universidad San Sebastian, Bellavista 7, Santiago 8420524, Chile

**Keywords:** nsPEF, NPS, nanopores, ionic channels, medical devices, cancer

## Abstract

Nanosecond Pulsed Electric Field (nsPEF) is an electrostimulation technique first developed in 1995; nsPEF requires the delivery of a series of pulses of high electric fields in the order of nanoseconds into biological tissues or cells. They primary effects in cells is the formation of membrane nanopores and the activation of ionic channels, leading to an incremental increase in cytoplasmic Ca${}^{2+}$ concentration, which triggers a signaling cascade producing a variety of effects: from apoptosis up to cell differentiation and proliferation. Further, nsPEF may affect organelles, making nsPEF a unique tool to manipulate and study cells. This technique is exploited in a broad spectrum of applications, such as: sterilization in the food industry, seed germination, anti-parasitic effects, wound healing, increased immune response, activation of neurons and myocites, cell proliferation, cellular phenotype manipulation, modulation of gene expression, and as a novel cancer treatment. This review thoroughly explores both nsPEF’s history and applications, with emphasis on the cellular effects from a biophysics perspective, highlighting the role of ionic channels as a mechanistic driver of the increase in cytoplasmic Ca${}^{2+}$ concentration.

## 1. A Brief History on the Development of Electric Pulses Technology

The use of electricity in humans can be traced back to the 18th century, when tissue damage was observed after the application of electric fields [1]. Despite the occurrence of lesions on the skin of humans and animals after exposure to electric sparks, the mechanism of action was far from being understood. Much latter, circa 1982, Neumann et al. achieved the first DNA transfection into cells [2] by applying a protocol including an electric field of 8 kV/cm for 5 μs, inducing a phenomenon in the cell membrane they termed *electroporation*. Almost a decade later, Pakhomov et al. demonstrated that the application of electric fields on cells creates water-filled lipid nanopores forming a stable, ion channel-like conduction pathway in the cell membrane [3]. Denoting its appropriateness, the definition of electroporation has remained intact for over 30 years: “electroporation is the transient loss of semi-permeability of cell membranes under the application of electric pulses, leading to ion leakage, the escape of metabolites, and increased cell-uptake of drugs, molecular probes, and DNA” [4]. Since its remote origins, this technology is nowadays widely used for several applications other than DNA transfection, such as electrochemotherapy [5,6], tissue ablation [7,8], extraction of chemical compounds [9,10], and microbial inactivation for food preservation [11], among others. The next significant step along the historical evolution of the application of electric pulses to biological systems occurred in 1995, when Schoenbach et al. developed a technique to generate high intensity nano-pulsed electric fields, on the order of 6.45 kV/cm with a duration ∼700 ns, to treat natural water used in industrial cooling systems [12]. This technique is nowadays known by the academic community either as nanosecond Pulsed Electric Field (nsPEF) or Nano Pulse Stimulation (NPS). Later on, Schoenbach started a longstanding collaboration with Stephen J. Beebe; together they pioneered the nsPEF field, studying systematically its effects in cells through both theoretical and experimental approaches, giving this technique a new spectrum of applicability (see Section 7). By stepping into the sub-nanosecond realm, inspired by a note from Carl E. Baum in 2005 and later published in 2007 [13], Heeren et al. used an impulse radiating antenna (IRA) instead of electrodes to deliver an electric pulse with a peak amplitude of about 250 kV and with a pulse-width of ∼600 ps [14]. This development added two main advantages to the field: the capability of delivering an electric pulse in the order of picoseconds, and the ability to target deeper body tissues, allowing the application of nsPEF in vivo. Figure 1 summarizes the main events through time in the development of nsPEF technology.

## 2. nsPEF and Ca^2^+-Mediated Apoptosis: An Evolutionary Prespective

Despite a long-dated and active debate, abundant literature suggests that one of the main cellular consequence due to the application of nsPEF in cells is the increase in the cytoplasmic concentration of Ca${}^{2+}$, impacting multiple cellular pathways (see Section 3).

Accumulated knowledge from molecular biology allows us to better understand the relationship between inner cell Ca${}^{2+}$ homeostasis and its role in the evolution of the eukaryotic cells [15]. A paleobiological hypothesis postulates that the prehistoric alkaline ocean contained extremely low Ca${}^{2+}$ concentration [16]: i.e., life emerged from a calcium-free medium (Figure 2). This hypothesis agrees with the fact that all life forms on Earth are supported by cells containing low cytoplasmic Ca${}^{2+}$ concentrations, a necessary condition to not perturb ATP metabolism since phosphate precipitates in the presence of Ca${}^{2+}$ [17]. Eukaryotic cells can achieve cytoplasmic nanomolar concentrations of Ca${}^{2+}$ by distributing it in organelles. These internal reservoirs allows a fast and focal release of Ca${}^{2+}$ at specific cell sites, generating a variety of cellular signals as a consequence of its reversible binding to calcium-binding proteins (CaBP) [18,19]. Thus, evolution took advantage of the increase of calcium in the internal media by turning calcium into a pleiotropic second messenger [20]. Calcium’s evolutionary protagonism is highlighted by an increase in the number of CaBP along evolution, which rises from nearly 70 in bacteria to more than 3600 in mammals [21,22], therefore improving the ability of eukaryotic cells to fine tune Ca${}^{2+}$ signals [23]. In this regard, nsPEF may represent a technological key suitable to open the Ca${}^{2+}$ Pandora’s box in cells, turning over the recalcitrant evolutionary path of life tending to exquisitely regulate the internal Ca${}^{2+}$ concentration and providing us with a unique tool to manipulate cellular metabolism.

Controlling cell homeostasis due to the rise in internal cytoplasmic Ca${}^{2+}$ concentration produces massive consequences to cell fate: from either proliferation and differentiation to apoptosis. Cases of cell proliferation induced by the application of nsPEF are scarce, and the underlying mechanism is still a matter of discussion. Using microalgae, Buchmann et al. hypothesized through proteomic analysis that this proliferation may be the result of the activation of some stress response pathway [24]. They found that two proteins were overexpressed after the application of nsPEF, one of them being the Na${}^{+}$/Ca${}^{2+}$ exchanger/integrin-β4. Integrins are related to growth stimulation as they signal guanine nucleotide-binding proteins [25]. The overexpression of these proteins agrees with the abiotic stress response in plants, which involves Ca${}^{2+}$ as an essential second messenger [26]. On the other hand, apoptosis is also triggered by nsPEF, a field with exponential growth given its application in cancer treatment [27]. Despite an abundance of experimental data, the exact underlying cellular mechanisms controlling this process are still a matter of debate. However, as mentioned before, available evidence shows that the primary effect of the application of nsPEF in cells is the sudden increase in cytoplasmic Ca${}^{2+}$ concentration [28,29].

## 3. nsPEF Action Mechanism: A Deep Controversy

Given the lack of an experimental setup able to follow cell changes on the nanosecond scale, and despite the substantial advances in the field during the past 15 years, the mechanism through which nsPEF increases cytosolic Ca${}^{2+}$ concentration is still a matter of discussion. Previous reports suggested that nsPEF produces similar effects in the cell membrane as those that occur with electroporation, albeit two main differences: the size of the pores induced in the membrane (termed nanopores) and their location [30,31,32]. Even though the induction of nanopores by nsPEF has not yet been experimentally confirmed, theoretical knowledge provides suitable foundations supporting this hypothesis. The application of an electric field with the necessary magnitude to reach voltage differences of one order of magnitude above the resting potential of the cytoplasmic membrane should be enough to transiently induce nanopores [33]. This is exactly the case in experimental nsPEF setups [34]. From the biophysical point of view, the electric field ($\vec{E}$) resulting from a voltage difference (ΔV) ($\vec{E}=\Delta V/d$, where *d* is membrane thickness) across the cell membrane generates a force over charged atoms ($\vec{F}=q\vec{E}$, where *q* is the charge) that may perturb membrane integrity. In fact, Vernier et al. observed from Molecular Dynamics (MD) simulations that pore formation was due to electrophoretic migration of charged phospholipids initiated by the field-driven alignment of water dipoles at the membrane interface [33]. This finding was further supported by experimental evidence showing that negatively charged phosphatidylserine migration from the internal membrane leaflet to the external one occurred as a result of the application of a nanosecond pulse above two MV/m [35,36]. Thus, available evidence suggest that the increase in cytoplasmic Ca${}^{2+}$ concentration produced by the application of nsPEF could be due to the formation of membrane nanopores. However, an important question remained: are these nanopores located in the plasmatic and/or the internal membranes? The first studies focused on answering this question suggested that the application of nsPEF indeed affects the internal membranes. Therefore, the increase in cytoplasmic Ca${}^{2+}$ concentration could be the result of this ion being released from internal organelles such as the sarcoplasmic reticulum [32,37,38]. This evidence provided an inflection point in cell manipulation: nsPEF was cataloged as the first non-invasive, drug-free technique affecting organelles without altering the cytoplasmic membrane [39,40].

Albeit slowly, since 2005, new data have tipped the evidence scale towards the recognition that the application of nsPEF may produce larger effects on the cytoplasmic membrane than on internal ones. Nowadays, a large body of evidence supports the notion that the application of nsPEF produces effects mostly on the plasma membranes, but not necessarily through nanopore formation. Table 1 briefly summarizes some studies focused on localizing nsPEF-induced nanopores. For further references regarding the formation of nanopores on the internal membranes, please see [32,41,42,43,44,45,46].

Despite its crucial role maintaining the integrity and fluidity of cellular membranes, and its being the most abundant molecule in biological membranes after phospholipids [54,55,56,57,58,59,60,61,62,63,64,65,66,67,68,69,70,71,72,73,74,75,76,77,78,79], the role of cholesterol during the formation of nanopores is poorly understood. Early experimental in vivo and in vitro studies, as well as theoretical approaches, were designed to explore the effects of the cholesterol composition of cellular membranes during nsPEF-induced electroporation [80,81,82,83,84]. Conductance analyses of electroporated membranes indicate that formed pores may have diameters between 0.9 nm and 10.6 nm depending on the applied current and ionic strength of the media [85,86]. On the other hand, the small polar head and large hydrophobic core of cholesterol decreases the probability of pore formation by diminishing cellular membrane conductance [87,88]. Trying to get insight of the structure and dynamics of nanopore formation using molecular simulations, nsPEF application was simulated by a constant electric field on a membrane bilayer. Using pure 1-palmitoyl-2-oleoyl-sn-glycero-3-phosphocholine (POPC) membranes, $\vec{E}$ = 0.2 V/nm was needed to induce a nanopore after 13.6 ns of simulation. In contrast, a higher electric field of $\vec{E}$ = 0.35 V/nm and a larger simulation time (33.5 ns) was requited to induce a nanopore when the authors used a membrane with POPC-50%M/cholesterol–50%M [84]. These data suggest that the presence of cholesterol in the membrane provides higher stability during the application of nsPEF. Of note, CHO-K1 cells having a cholesterol-depleted membrane by using methyl-β–cyclodextrin (MβCD) were more sensitive to nanopore formation after the application of nsPEF pulses (10–150 kV/cm, 10–600 ns, up to 150 pulses) [89]. In [90], using the same nsPEF protocol and cell line as before, the authors used 3-(4,5-dimethylthiazol-2-yl)-2,5-diphenyl tetrazolium bromide (MTT) to evaluate toxicity. A linear relationship between the sensitivity to nsPEF to the amount of cholesterol removed with MβCD, and the incorporation of MTT was found [89,90], denoting the importance of cholesterol to maintain the structure and integrity of cell membranes under nsPEF application.

## 4. Dissecting the Biophysical Principles behind nsPEF’s Effects

In general terms, nsPEF can be classified as non-invasive electrostimulation [39,40]. Its application does not involve the absorption of energy by molecules, except for kinetic energy, as is the case of standard ionizing radiation techniques such as X-rays, NMR, PET, and cancer radiotherapy. As mentioned before, the main effect of the application of nsPEF into cells is the movement of charged species under the influence of the force resulting from the potential difference across the cell membrane. Therefore, this phenomenon can be classified between electrostatic and electrodynamics, because the nanometric timescale of the applied pulse creates an electric field that is changing in time over the cell membrane. An excellent review by Schoenbach et al. dives deeper into these matters [91].

Despite the lack of an explanatory consensus regarding how nsPEF affects either the plasmatic and/or the internal membranes, a common knowledge base has accumulated indicating that the location of nsPEF’s effects could be related to its intensity and time interval and the characteristic membrane charging time. It is important to address that the $\vec{E}$ delivered by an nsPEF protocol would not instantaneously increase the membrane $\vec{E}$. As in any RC circuit, a capacitor (i.e., the cytoplasmic membrane) does not fully charge until a certain time lapses, which is related to the RC time-constant (τ=RC, where *R* is the resistance and *C* the capacitor’s capacitance). In the comparative case of a cell, R represents the value given by the cell’s surrounding medium. The equation describing the capacitor’s voltage increment over time in an RC circuit is:(1)$$V\left(t\right)=V_{0}e^{-t/\tau }$$
where V(t) is the capacitor voltage at a certain time *t*, $V_{0}$ is the capacitor voltage at time t=0, and t=τ is the time when the capacitor reaches around 63% of its charge capacity, almost reaching its maximum capacity around 4τ.

Theoretical approaches available in the literature have described the increase in membrane voltage as a function of time, and the dependency of the membrane time-constant ($\tau_{m}$) with the surrounding media [92] as described by:(2)$$V_{g}\left(t\right)=1.5aE_{0}cos\theta \left(1-e^{-t/\tau_{m}}\right)$$
where $V_{g}\left(t\right)$ is the voltage difference across the cytoplasmic membrane of a spherical cell, *a* is the cell radius, θ is the polar angle measured with respect to the electric field ${\vec{E}}_{0}$, and $\tau_{m}$ is the relaxation time-constant [49] which is also known as membrane charge time, charging time-constant [35,47,50,93,94], or charging time [27,48,50]. It is well known that Equation (Equation 2) describes exponential growth for $V_{g}\left(t\right)$, with a limit of $1.5aE_{0}cos\theta$.

On the other hand, the constant $\tau_{m}$ can be defined by Equation (Equation 3) as follows:(3)$$\tau_{m}=aC_{m}\left(1/2\sigma_{e}+1/\sigma_{i}\right)$$
where $C_{m}$ is the membrane capacitance per unit of area, $\sigma_{e}$ is the external conductivity, and $\sigma_{i}$ is the internal conductivity. In mammalian cells, $\tau_{m}$ is characterized around ∼100 ns [35,95]. Of note, $\tau_{m}$ does not represent the membrane charge time; it is the time when the membrane reaches 63% of its charge capacity, reaching 95% in 3τ [93]. As noted, a controversy arises when considering that the application of nsPEF protocols using time intervals far below the membrane $\tau_{m}$ are also capable of producing nanopores. Moreover, as seen in Table 1, no clear relationship between the intensity, duration, and the area of impact in the cell can be established.

A theoretical analysis could shed some lights on this controversy; either the application of nsPEF affects the plasmatic membrane, the inner organelle membranes, or both. If the cell is considered, for the sake of simplicity, as a solid metal and conducting sphere, the electrons contained in the sphere should migrate to the anode when an external electric field is applied. After a characteristic amount of time, this continuous migration of electrons should result in an asymmetric charge distribution, creating a self-induced electric field around the sphere (the reaction field) that could nullify the external electric field, resulting in a zero electric field inside the sphere. A similar phenomenon may occur in cells due to the application of nsPEF, but instead of electron movement, there are ions moving around creating equilibrium in the charge distribution to be reached in a much longer time (Figure 3). As the characteristic time to nullify the external electric field in cells is in the order of microseconds or even milliseconds [96] during standard electroporation, where pulses last longer, the reaction field in the cells should equilibrate, and charge relocation should cease. This is not the case when nsPEF is applied because the pulse duration is in the nanosecond scale, and therefore the movement of charges is not able to reach the necessary equilibrium so as to nullify the applied external electric field. Hence, internal charges will continue to move by the influence of the external electric field induced by the application of the nsPEF, continuously perturbing the structure and dynamics of internal structures in the cell. Consequently, with this analysis, long-lasting nsPEF protocols will eventually perturb not only the internal structures of the cell but also the plasma membrane, as can be seen in Table 1.

## 5. It’s All about Pores? In the Shade of Voltage-Gated Ion Channels

### 5.1. Voltage-Gated Channel (VGC) Activation Mechanism

Despite the abundant literature suggesting that the primary effect of the application of nsPEF protocols in cells should be the formation of nanopores, available evidence indicates that the activation of voltage-gated ion channels is also a relevant effect (see Section 5). However, it is also important to recognize the controversy arising from the significant differences between the time scales of the application of nsPEF protocols, in the order of nanoseconds, and the characteristic activation time of ion channels, in the order of ms [97,98]. To further understand the implications behind this time scale controversy, it is first necessary to explore with greater details the activation mechanisms of VGCs. During VGC activation, displacement of the charges tethered to the Voltage Sensing Domain (VSD) gives rise to transient gating currents. Kinetics indicate that during VGC activation, the VSD undergoes a complex conformational change that encompasses many transitions [99,100,101]. Four main models have been proposed to rationalize the transfer of charge during VGC activation, all of them associated with the motion of the S4 helix [102,103]. These four models are called the *helical screw-sliding* model [104,105], *kinetic* model [106], *paddle* model [107] introduced following the publication of the K${}^{+}$ channel (KvAP) structure [108], and *transport* model [109].

***Helical screw-sliding model***: This simple model proposes that the S4 helix is responsible for maintaining the pore in a closed state during the resting potential. This is achieved by displacement of the positively charged S4 helix attracted by negative charges close to the cytoplasm. During depolarization, this attraction would vanish, and the system would return to a 60${}^{\circ }$ rotation of S4 around its geometric axis. This rotation is accompanied by a vertical displacement of 5 Å to the extracellular side. According to this model, the positive residues of the S4 helix form salt bridges with acidic residues on opposite transmembrane segments. This model was based on the sodium channel transmembrane structure determined for an *Electrophorus electricus* channel [110,111]. Charge reversal mutagenesis [112] and disulfide linking [113] were used to probe charge interactions within the VSD of different VGCs. These works demonstrated the existence of a sequential ion pair formation involving S4 basic residues, typical on this type of channels. These interactions were key to conformational changes of the VSD upon VGC voltage activation. Subsequent works demonstrated similar key interactions required to characterize VGC activation in a more detailed fashion. Using site mutagenesis, two negatively-charged residues and a highly conserved one were identified as “catalyzers” of the transfer of each of the VSD basic residues across the membrane electric field [114]. This cluster of residues is known as the *Charge Transfer Center*.***Kinetic model***: In this model, at hyperpolarizing potentials, the basic amino acid residues of S4 are connected with an intracellular water crevice, maintaining the channel in a closed state. Upon depolarization, the S4 helix tilts and rotates 180${}^{\circ }$ around its geometric axis, allowing it to be connected to an extracellular water crevice. This conformational change of the S4 helix pulls the intracellular side of the S5 transmembrane helix, leading to a rotation and pulling of the intracellular section of the S6 helix, which forms the pore, opening the channel. This model was proposed for mammalian ion channels based on the gating mechanism of the prokaryotic KcSa potassium channel [106,115,116].***Paddle model***: The paddle in this model circumscribes to the helix-turn-helix motif between the S3 and S4 helices. The paddle moves its center of mass nearly 20 Å and tilts towards a more vertical orientation. Since each paddle in the four VSDs of the VGC contains four arginine residues, with one electron charge unit per arginine [117,118], their displacement would account for the total gating charge in the Shaker K${}^{+}$ channel of 12–14 electrons (3.0–3.5 electrons per subunit) [117,118,119]. Ionic interaction with S2 and S3 helices would stabilize the movement of paddle charges. The S4–S5 linker is pulled to open the VGC pore as a result of the paddle movement.***Transport model***: Experimental observations on the Shaker K${}^{+}$ channel with fluorescent resonance energy transfer (FRET) concluded that the S4 helix does not move during channel activation. To explain this observation, the same author of the kinetic model proposed that the voltage gating is due to a transmembrane field rearrangement. In this rearrangement, VSD’s water crevice plays a key role. In this model, the S4 helix gyrates 45${}^{\circ }$ around its geometric axis and has a vertical shift of less than 2 Å. The S4 turn relocates the S4 charges, reverberating from a deep, internally facing aqueous crevice in the closed state to an external water crevice when it opens. This model is supported by experimental evidence of a proton-conducting pore in a mutant Shaker channel in the closed state [120], the strong dependence of gating charge quantity on intracellular ionic strength, and the measurement of an amplified membrane electric field near the second gating-charge amino acid residue [121].

### 5.2. The Time-Scale Controversy behind Ion Channel Activation by nsPEF Protocols: The Role of MD Simulations

As noted in the previous section, regardless of the actual mechanism for the activation of VGCs, the extensive conformational changes occurring during the activation of VGCs require elapsed times in the order of ms [97,98]. Therefore, new data are needed to provide a biophysically sound explanation for the activation of ion channels in the nanosecond time scale, as occurs with nsPEF. For this task, molecular modeling and MD emerge as suitable tools to provide an atom-based description of the structural and dynamical changes occurring in ion channels under the application of nsPEF protocols.

MD results strongly suggest that the conformational changes at the VSD proceed after the ion channel closes, providing new evidence to support the *kinetic model* [122]. Recently released X-ray structures of ion channels also support the *kinetic model* through the observation that their VSD structures in the active state of the channels are linked to a closed pore domain [123,124]. Moreover, MD results able to reach a resting state of the channels exhibit a VSD conformation that is in agreement with the *kinetic model* [125,126,127,128,129,130]. Despite the agreement between these results, the large negative voltages used during the MD protocols must be taken with caution because there are neither in vivo nor in vitro experiments performed under the same conditions.

Another observation from long MD simulations is that the pore domain has to undergo a de-wetting process of its intracellular water crevices before being able to reach a closed state. However, experimental evidence contradicts this observation, since the *Shaker* K${}^{+}$ channel in the closed state may still carry solvated ions in its pore cavity [131,132,133]. Furthermore, inactivated ionic channels also contain water-filled crevices in their closed pores [123,124,134].

There are different approaches to simulate the application of an external electric field over the membrane akin to nsPEF protocols while running MD protocols. The most common one is introducing a uniform electric field $\vec{E}$ perpendicular to the membrane plane throughout the entire simulation box. This gives rise to a force $\vec{F}=q_{i}\vec{E}$ that is applied to all charges $q_{i}$ in the simulation. The value of the transmembrane voltage (TMV) will be $\Delta V_{m}=EL_{Z}$, where $L_{Z}$ is the length of the *Z*-axis of the simulation Box. In order to avoid the accumulation of ions when the external electric field is applied, it is recommended that this method be used in the absence of salts. It is important to address that this electric field implementation has a certain appearance of artificiality that can cause some concerns [135,136], mainly because the force over charged atoms is independent of their position.

A more realistic way to reproduce the TMV is through the method of imbalancing ions. In fact, the in vivo TMV is caused by a small charge imbalance across the membrane [137,138]. To mimic this imbalance, there are two different implementations. The first one, known as the two-membranes setup, is achieved by using a twin phospholipid bilayer system that includes two independent bulk phases with unequal ion distributions [135]. Despite being suitable to produce a TMV as a consequence of the ion imbalance, this protocol significantly increases the number of atoms required for an MD simulation, scaling up the computational cost of the simulation. To bypass this problem, a second implementation was proposed, consisting of a single bilayer and an air–solvent interface, that also results in two independent bulk phases [136]. Despite being more efficient than the double-membrane method, this protocol may produce unwanted surface phenomena at the air–solvent interface.

### 5.3. Voltage-Gated Calcium Channel

Most of the available evidence suggests that the increase of cytoplasmic Ca${}^{2+}$ concentration is the result of nanopore formation at the cytoplasmic membrane. Back in 2002, Beebe et al. were the first to propose that ion channels could be possible targets of nsPEFs protocols [139]. Since that time, ion channels have gained protagonism in the field due to strong experimental evidence (see below). Intuitively, due to their sensitivity to changes in transmembrane potential and due to their ability to transport Ca${}^{2+}$, the main target of nsPEF protocols should be Voltage-Gated Calcium Channels (VGCC).

VGCCs fall into two major categories: high-voltage-activated (HVA) channels that open in response to large changes in voltage across the cell membrane, and low-voltage-activated (LVA) channels, which are activated by small voltage changes [140,141] close to the typical resting membrane potential of neurons (∼80 mV). Based on biochemical and molecular analyses [142], HVA channels have been characterized as heteromultimeric protein complexes formed through the co-assembly of a pore-forming $\alpha_{1}$ subunit, having ancillary $\alpha_{2}\gamma$, β, and γ subunits, whereas LVA channels appear to lack the latter. The $\alpha_{1}$ subunit is the key determinant of calcium channel subtypes. There are three major families of $\alpha_{1}$ subunits (termed Ca${}_{v}$1, Ca${}_{v}$2, and Ca${}_{v}$3), each consisting of several members [143]. The Ca${}_{v}$1 channel family encodes three different neuronal L-type channels (termed Ca${}_{v}$1.2, Ca${}_{v}$1.3, and Ca${}_{v}$1.4) plus a skeletal-muscle-specific isoform, Ca${}_{v}$1.1 [144,145,146,147]. These channels are sensitive to a number of different dihydropyridine (DHP) antagonists and agonists [148]. The Ca${}_{v}$2 channel family includes three members (Ca${}_{v}$2.1, Ca${}_{v}$2.2, and Ca${}_{v}$2.3). Through alternative splicing and assembly with specific ancillary subunits, Ca${}_{v}$2.1 gives rise to P- and Q-type channels [149,150], which are both blocked (albeit with different affinities) by ω–agatoxin IVA, a peptide isolated from spider venom [151]. Ca${}_{v}$2.2 encodes N-type channels [147,152] that are selectively inhibited by ω–conotoxins and the GVIA and MVIIA toxins isolated from mollusk venom. Ca${}_{v}$2.3 corresponds to R-type channels [153] that can be inhibited by SNX-482, a peptide present in tarantula venom [154,155]. There are three types of Ca${}_{v}$3 channels (Ca${}_{v}$3.1, Ca${}_{v}$3.2, and Ca${}_{v}$3.3), all of which represent T-type calcium channels [156,157,158]. Ca${}_{v}$3 channels can be distinguished by their sensitivity to nickel and relative resistance to blocking by cadmium ions, which block all HVA channels in the low micromolar range (for review, see [159]).

All ten α subunits share a common topology of four major transmembrane domains, each of them containing six membrane-spanning helices, termed S1 to S6. Helices S1 to S4 form the VSD, including the positively charged S4 segment, the key that controls voltage-dependent activation [160]. In addition, they have a typical re-entrant P loop motif between S5 and S6 that forms the permeation pathway (Figure 4). Each of the P loop regions contains highly conserved negatively charged amino acid residues (glutamate in HVA channels) that cooperate to form a pore that is highly selective for permeable cations such as calcium [161,162,163], barium, and strontium [164] and that interact with non-permeable divalent cations such as cadmium [165].

The majority of the structural/functional information about VGCC has been deduced from site-directed mutagenesis and generation of chimeric calcium channel subunits. Unlike potassium and bacterial sodium channels, it has not yet been possible to obtain crystallographic structural information concerning entire mammalian VGCC subunits, although structures of the $\alpha_{2}\gamma$ subunit bound to a fragment of the $\alpha_{1}$ subunit I-II linker have been resolved by multiple groups [166,167]. Furthermore, co-crystallographic studies and even NMR structures of calmodulin bound to Ca${}_{v}$1.2 and Ca${}_{v}$2.1 have been reported [168,169,170,171,172,173,174]. Cryo-EM structures have revealed crude structural information about this channel subtype [175,176,177,178]; however, they do not have enough resolution to gain insight into the structural basis of channel function. Based on the crystallographic structures of potassium channels released in 2005 [179], several homology models of α subunits have been constructed and used to model drug interactions, in particular with L-type channels [180,181,182]. While these works have provided some advances in our understanding of subunit regulation of VGCC, it remains to be determined whether the observed interactions are relevant to actual conformations in holochannels or perhaps modified by the presence of transmembrane regions and other intracellular domains. These studies about VGCC using the structures of other ion channels are supported by the fact that the fourth VSD subunit is ubiquitous to VGC [183] since the main differences in selectivity arise from the S6 transmembrane helix that forms the pore domain.

**Figure 4 ijms-23-06158-f004:**
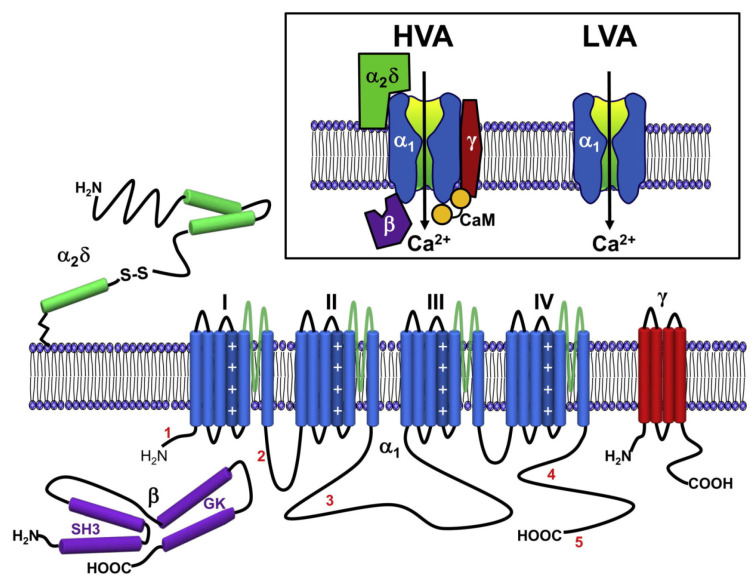
Representation of the membrane topology and secondary structure of HVA channels. In blue: the common structure of HVA and LVA channels, with their four VSD (I, II, III, and IV). The dark blue cylinder in each VSD is the charged S4 helix. The single blue cylinder in each VSD is the S6 helix that forms the pore domain. In green, purple, and red: the other subunits of HVA channel. The lengths of lines correspond approximately to the lengths of the polypeptide segments represented. Inbox at top: comparative representation of the structure of HVA and LVA calcium channels ($\alpha_{1}$) and the subunits of HVA channels ($\alpha_{2}\delta$, β, and γ). Taken from the journal *Neuron*, article “Neuronal Voltage-Gated Calcium Channels: Structure, Function, and Dysfunction” [184].

Now that we have explored the topology, structure, and function of VGCC, we will focus on the literature suggesting that the application of nsPEF protocols may induce the activation of these channels. Rogers et al. in 2004 were the first group to stimulate isolated muscle fibers with an nsPEF (∼1 ns, 24 kV/cm). They observed muscle contraction induced by the application of the nsPEF protocol and that the duration of the strength curve extended linearly up to 1 ns. According to these authors, these data suggest the activation of some ionic channel without electroporation. Moreover, they suggested that the most probable cause should be the increase of intracellular Ca${}^{2+}$ concentration via VGCC activation [185]. Later on in 2010, Craviso et al. used bovine cromaffin cells to apply an nsPEF protocol consisting of 50 kV/cm with a pulse cycle of 5 ns to show that the entry of Ca${}^{2+}$ is mainly through L-type VGCC. Consistently, ω–conotoxin GVIA (N-type VGCC blocker), ω–agatoxin IVA (P/Q-type VGCC blocker), and ω–conotoxin MVIIC (N/P/Q-type VGCC blocker) reduced the increase in cytoplasmic Ca${}^{2+}$. Moreover, the simultaneous blockage of L-, N-, and P/Q-type channels by using a cocktail of VGCC inhibitors completely abolished Ca${}^{2+}$ entry. These results suggest that the increase in cytoplasmic Ca${}^{2+}$ occurs only through VGCC [186]. More recently, by using the human glioblastoma U87 MG cell line, Burke et al. demonstrated the role of VGCC, both L and T-type, in Ca${}^{2+}$ influx due to nsPEFs. On top of that, these authors also suggested that other ionic channels could be involved, such as the Ca${}^{2+}$-gated BK potassium channel and the TRPM8 (transient receptor potential) channel [187]. It is worth mentioning that Ca${}^{2+}$ channels were not uniquely identified as a target of the application of nsPEF protocols. Another ion channel more recently described as a nsPEF target is the Na${}^{+}$ voltage-gated channel (VGNC) [186,188]. These findings are of particular interest for the manipulation of excitable cells through nsPEF protocols: they can trigger action potentials in excitable cells by the activation of this channel. Thus, findings relating VGNC activation by nsPEF are exciting and deserve more attention. Table 2 was prepared to summarize the latest findings related to VGC activation by the application of nsPEF protocols, including VGNCs.

## 6. Protein-Mediated Electroporation: An Additional nsPEF Effect?

As discussed above, there is plenty of evidence supporting the notion that lipidic nanopores are formed in the internal cellular membranes due to the application of an nsPEF protocol. Moreover, theoretical approaches also support this notion, leading to the conclusion that some electroporation protocols may change the permeability of cellular organelles [192]. Therefore, the definition of electroporation should be extended to include transient changes in the semi-permeability of both cytoplasmic and internal membranes. It is well known that the transient loss of semi-permeability of cellular membranes by an external electric stimulus is not only due to the formation of aqueous lipidic pores but may also occur by a broader range of biophysical and biochemical mechanisms, ranging from pH changes, the use of chaotropic agents, and ion imbalance [193]. For this reason, the term *electropermeabilization* has being coined in the field to refer to changes in membrane permeabilization, not necessarily due to the formation of lipidic pores. By contrast, the term electroporation should strictly be used to refer to changes in the membrane solely as a result of the formation of aqueous lipidic pores [193]. As a consequence, we will discuss additional electropermeabilization processes that may occur due to the application of nsPEF protocols. To do so, we will focus on the role of transmembrane proteins other than ion channels (discussed in Section 5). The first association between transmembrane proteins and electropermeabilization processes occurred in 1980: the application of an external electric stimulus in erythrocytes produced an incremental increase in the electrical conductivity of the transmembrane Na${}^{+}$/K${}^{+}$-ATPases [194]. Ten years had to pass for a scientifically sound explanation of this phenomenon: the external electrical stimuli would generate sufficient heat over the protein for its denaturation, permitting the passage of ions, generating the measured macroscopic current [4,195].

The application of MD protocols has further expanded the mechanistic comprehension of the electropermeabilization process that could be mediated by transmembrane proteins. Recent evidence coming from the application of external stimuli mimicking the application of nsPEF in MD simulations points towards the formation of pores in transmembrane proteins. The first work studying pore formation in transmembrane proteins by the application of an electric stimulus using MD simulation was published in 2018 [196]. In this study, an intraprotein electropore persisting more than ∼50 ns was produced in a human aquaporin when the simulation box was subjected to a continuous electric field of 0.2 V/nm. Of note, this electropore was resealed within ∼20 ns after turning off the external electric field. By performing an MD protocol, Rems et al. (2020) [197] registered the formation of either simple or complex pores located in the VSD in three distinct VGCs: a bacterial VGNC, a eukaryotic VGNC, and a human hyperpolarization-activated cyclic nucleotide-gated channel. Complex pores are produced by massive rearrangements of transmembrane segments of the affected channels and have been proposed as the source of the theorized lipidic pores that may be stabilized by the presence of ions and other parts of the channels, such as TMHs [198,199,200]. When the three aforementioned VGCs are subjected to an oscillating hyperpolarizing/depolarizing TMV of ±1.5 V for 600 ns, the formation of different protein pores is promoted. Interestingly, Rems et al. discuss that the formation of simple and complex pores at the VSD can explain some experimental observations: (i) major VSD rearrangements are expected to turn the VSD dysfunctional, disrupting the gating of VGCs. This phenomenon could explain electrophysiological measurements showing that electroporative submicrosecond electric pulses can decrease ionic currents through VGNC and VGCC channels in different excitable cells [201,202,203]; (ii) some VSDs are easier to be porated than others, which may explain why some channels appear to be affected at weaker electric fields than others [203]; (iii) structurally different pores forming in the VSDs at either hyperpolarizing or depolarizing TMV may explain why the decrease in channel conductance depends on the polarity of the TMV [204]; (iv) observed complex pores remained open during a 1 μs simulation, suggesting that, at least during that timeframe, the VSD would not spontaneously refold back. These complex pores observed in MD simulation, lasting from tens to hundreds of nanoseconds [205,206], offer an interesting point of view to the discussion of why cell membranes remain permeable for seconds or minutes after application of nsPEF protocols [193].

Recently, our group published an MD simulation article in which we also observed the formation of complex pores in a VSD, in this case belonging to a human VGCC [207]. These pores were created by the application of an external electric field of 0.2 V/nm for 50 ns (mimicking an nsPEF) using a cellular membrane prototype containing POPC and cholesterol in a 1:3 ratio. Reformulating the topic of this section as the capability of nsPEF to induce pores in transmembrane proteins, we may argue that in the available literature there is robust data obtained from MD simulations supporting a positive answer. However, to fully address this question, experimental data are mandatory. We strongly support Rems et al. [197] exhortation to experimentalists for further investigation in this issue. We certainly agree that eventual biotechnological applications for such an nsPEF device capable of forming protein pores are highly stimulating to the imagination. For further insights on the role of MD simulations to study the effects of nsPEF protocols, in particular conformational changes occurring on the kinesin nanomotor and other proteins, please refer to [208,209], respectively.

## 7. nsPEF Applications

In the following section, we briefly review some applications of nsPEF technology. Despite still being under development, there are interesting perspectives regarding the development of standard nsPEF devices that may be widely used in the near future, particulary in health-related applications. As mentioned before, nsPEF is a versatile, non-invasive, and cheap technology that can manipulate cellular membranes and even transmembrane proteins with an exquisite fine-tuning.

### 7.1. In Human Health

**Activation of excitable cells**:**Cardiac cells**: nsPEF (10–80 kV/cm, 4 ns, 1–20 pulses with 200/400/600 ms intervals) can indirectly lead to cardiac cell excitation. Of note, these results challenge the concept of chronaxie: minimum time required for an electric current to double the strength of the rheobase in order to stimulate a muscle or a neuron. The use of nsPEF technology to excite cardiac cells and mobilize intracellular Ca${}^{2+}$ may prove valuable for cardiac pacing and defibrillation [210]. For other related studies see [211,212,213].**Neurons**: nsPEF (27.8 kV/cm, 10 ns, single pulse) was sufficient to initiate action potentials. The observed effect was repeatable and stable. These results highlight the potential use of ultrashort pulsed electric fields for stimulation of subcortical structures and suggest they may be used as a wireless alternative for deep brain stimulation [214]. For other related studies see [188,215,216,217].**Phenotype manipulation**:**Differentiation**: nsPEF (1.5–25 kV/cm, 300 ns, 5 pulses) can induce proliferation and myotubule maturation or nodule formation in myoblasts and osteoblasts, respectively. Myoblasts were isolated from hind-limb skeletal muscle of four-week-old mice *Pten*${}^{MKO}$, and primary human osteoblasts were obtained from a vendor (Sciencell^®^) [218].**Dedifferentiation**: nsPEF (10–20 kV/cm, 100 ns pulse) induces dedifferentiation partially through transient activation of the wnt/β–catenin signaling pathway in porcine chondrocytes [219].**Gene expression**: nsPEF (20 kVcm, 80 ns, various combinations of pulses) dramatically elevated c-Jun and c-Fos mRNA levels, which correlated with the observation of c-Jun N-terminal kinase (JNK) pathway activation in HeLa S3 [220]. For related studies see [219,221,222,223,224].**ntiparasitic**: Cystic echinococcosis is a widely endemic helminthic disease caused by infection with metacestodes (larval stage) of the *Echinococcus granulosus* tapeworm. Application of nsPEF (21 KV/cm, 300 ns, 100 pulses) caused a significant increase in the death rate of protoscolices (future heads of the adult worms) [225]. For related studies see [226,227].**Wound healing**: nsPEF (30 kV/cm, 300 ns) induced platelet rich plasma aggregation and platelet gel formation. These gels are applied to soft and hard tissue wounds, where they enhance healing [30]. For other related studies see [228,229,230].**Immune response**: Using in vivo experiments, nsPEF (15 kV, 100 ns, 400 pulses) induced translocation of calreticulin in rat tumor cell-surfaces, a molecular pattern associated with damage that is indicative of immunogenic cell death (ICD). The nsPEF also triggered CD8-dependent inhibition of secondary tumor growth, concluded by comparing the tumor size using rats depleted of CD8${}^{+}$ cytotoxic T-cells under the same nsPEF treatment. The first group showed an average size of only 3% of the primary tumor size compared with the 54% shown by the CD8${}^{+}$-depleted rats. Additionally, with immunohistochemistry it was observed that CD8${}^{+}$ T-cells were highly enriched in the first group. Furthermore, it was shown that vaccinating rats with isogenic tumor cells (MCA205 fibrosarcoma cell line) treated with nsPEF (50 kV, 100 ns, 500 pulses) stimulates an immune response that inhibits the growth of secondary tumors in a CD8${}^{+}$-dependent manner [231]. This work opens the door to the fabrication of cell-based vaccines using nsPEF stimulation to promote an improved immune response. For other related studies reporting tumor ablation through an antitumor immune response using nsPEF see [232,233,234,235,236].**Cancer**: This is by far the most-studied nsPEF application, with 46 in vitro studies up to 2016 [27] and over 100 so far. Recently, preclinical animal studies have demonstrated that nsPEF can induce local and systemic CD8${}^{+}$ T-cell mediated adaptive immune response against tumors [233,236]. In clinical trials, nsPEF proved to be a safe and effective therapy against basal cell carcinoma [237,238]. There are other novel techniques to combat cancer that also use electric fields, known as electrochemotherapy [239,240], irreversible electroporation [7], and electro-gene therapy [7]. Electrochemotherapy and electro-gene therapy use electroporation to achieve the anti-tumoral effect of other agents. In irreversible electroporation, cytoplasmic membranes of tumor cells cannot recover from permeabilization, causing cell death mainly by necrosis. Unlike the just mentioned electro-technique, nsPEF is cell-dependent. A possible explanation for this may be related to apoptosis (programmed cell death type 1 [241]), which is a tightly controlled cell process and different in each cell type [242]. Thus, if nsPEF induces apoptosis, as seems to be the case, it is expected to exhibit cell-dependent responses. This makes nsPEF an extraordinary tool, with specific responses based on tuning the intensity, duration, and number of pulses. There are several examples of cell dependence and nsPEF. Stacey et al. in 2002 demonstrated that exposing cancer cells to nsPEF with 60 kV/cm could induce DNA damage [243] (Figure 5). Beebe et al. in 2002 studied the antitumor effects of nsPEF on Jurkat cells, with pulses at 60, 150, and 300 kV/cm [139]. Xinh ua Chen et al. in 2012 applied nsPEF with 900 pulses at 68 kV/cm to ablate hepatocellular carcinoma [244]. Nuccitelli et al. in 2013 inhibited human pancreatic carcinoma using 100 pulses of 100 ns duration and 30 kV/cm [245]. More importantly for nsPEF as cancer treatment, tumor cells are more sensitive to nsPEF than normal cells [246]. See Figure 6 for an example of a nsPEF device suitable for use in cancer treatment.

### 7.2. Industrial

**Cell proliferation**: nsPEF (10 kV/cm, 100 ns) can increase *Arthrospira platensis* SAG 21.99 (a cyanobacteria) cell growth after repeated pulses in the exponential growth phase. The effect was most pronounced five days after treatment. Treatments with nsPEF might improve sustainable and economical microalgae-based biorefineries [24]. For other studies see [218,248,249].**Fermentation industry**: nsPEF (15 kV/cm, 100 ns, 20 pulse) increased avermectin (anthelmintic and insecticidal agent) production in *Streptomyces avermitilis* by 42% and reduced the time needed for reaching a plateau in the fermentation process from 5 to 7 days [250]. For other related studies see [251].**Food industry**: Microalgae are a novel food ingredient of increasing interest as they can be grown on non-arable lands and fixates CO${}_{2}$ when grown photoautotrophically. Treatment with nsPEF (5–100 kV/cm, 2–100 ns) reduced total bacterial contamination >log${}_{10}$ in *Chlorella vulgaris* cultures without compromising the microalgae. For related studies see [252,253].**Seed germination**: nsPEF (10–30 kV/cm, 100 ns, 20 pulses) application significantly affected seed germination and pre-growth of *Haloxylon ammodendron* (Figure 7). This is probably due to the exogenous and endogenous NO generated in the nsPEF seed-treatment system [254]. For related studies see [255,256].

## 8. Challenges and Future Perspectives of nsPEF’s Effect on Cells

As a relatively new technology (just 25 years old), the accelerated development of nsPEF comes with a series of challenges. While some are the lack of experimental setups to follow changes in membranes at the nanosecond time scale, others are related to the lack of experimental evidence supporting the formation of nanopores in proteins. On top of these, the existence of contradictory results related to both the temporal scale and the actual target (as discussed in previous sections) are a matter of active debate in the community. However, being the first drug-free, non-ionizing technology directly affecting cellular organelles, nsPEF opens a biotechnological Pandora’s box potentially enabling exciting new applications in a variety of fields. Therefore, a compendium of both experimental and theoretical data are needed in order to promote a better understanding of this extraordinary phenomenon. Focusing on this aim, in the following section we offer a brief discussion of some other relevant topics surrounding this amazing technology.

### 8.1. Nomenclature, Abbreviations, and Mathematical Formulas

While some authors refer to this technology as “Nanopulse Stimulation (NPS)”, others may use “Nanosecond Pulsed Electric Field (nsPEF)”. Thus, using only one of these nomenclatures and/or abbreviations while searching the literature may lead to missing some valuable research. It is worth mentioning that nsPEF seems to be a much better term than NPS because the latter is widely used in other fields to refer to: nanoparticles (NPs) in nanotechnology [257,258]; noise-power spectrum (NPS) in electronics and signal analysis [259,260]; and net promoter score (NPS) in economics and customer care [261,262].

A brief but important mathematical formalism: the parameter $\tau_{m}$ may lead to some confusion (see Section 4). This symbol represents the membrane relaxation time-constant (for cytoplasmic or internal membranes) and not the charging time of the membrane. Keeping this difference in mind is important because their confusion may affect the outcome of experimental protocols. For instance, to produce an effect on internal membranes rather than the cytoplasmic membrane, a pulse duration below the membrane charging time should be used. It is important to remark that to achieve 95% of the charging capacity of the membrane, ∼3$\tau_{m}$ time should be elapsed. Therefore, in order to affect mainly the internal membranes, a pulse duration below ∼3$\tau_{m}$ should be used.

It has also been recommended by several authors not to assume that every cell line has a $\tau_{m}$ near 100 ns, as is described in some articles as an approximate value of $\tau_{m}$ for mammalian cells. Theoretical approaches postulate that $\tau_{m}$ is directly proportional to the cell radius and has a strong dependency on the cytoplasmic conductivity and cell medium conductivity (Equation (Equation 3)). Thus, $\tau_{m}$ is a cell-dependent value not only influenced by the size but also by the inner ionic strength and that can be modulated by changing the medium conductivity. Taking all of this information into account, when choosing an nsPEF protocol to accomplish a desired cell effect, it is highly recommended to consider cell size, membrane composition, and medium conductivity.

### 8.2. Nanopores

As mentioned before, there is a lack of experimental evidence demonstrating the formation of nanopores. However, theoretical data, mainly coming from MD simulations, and indirect experimental evidence suggest the formation of these structures as a primary nsPEF effect. Nevertheless, as discussed in Section 3, their exact localization, either on the cytoplasmic membrane or internal membranes, is still a matter of debate. The actual capability of internal membranes to be perturbed by an external electric field due to the application of nsPEF depends, as discussed in Section 4, mainly on the time and intensity of exposure. If the exposition time is larger than that of the charging time of the cytoplasmic membrane, then the electric field in the interior of the cell will be nullified (Figure 3) and any nanopore formation should be neglected in the internal membranes. That being said, many of the analyses of classic articles in the field speculate precipitation effects. It is expected that during the application of nsPEF, before the cytoplasmic membrane charge time is achieved, the cell interior is actually exposed to the electric field. Hence, an inner ionic current could be induced by the movement of charges, making the membrane voltage difference large enough to induce nanopore formation in internal membranes. These internal nanopores could play an important role to better explain the nsPEF effect, particularly when contrasted with the classic view allowing the formation of nanopores exclusively on the cytoplasmic membrane. Of note, the internal nanopore hypothesis is supported by evidence pointing towards the lifetime of nanopores varying from nanoseconds up to 1 s, according to early results [35,45,95,263,264,265]. Moreover, recent results increase the duration of nanopores even to the order of minutes, a timeframe where cells exposed to cytoplasmic membrane pores will collapse due to osmotic shock or will undergo apoptosis or necrosis depending on the cellular pathways activated [3,52,53,266,267,268,269].

#### Nanopores, Cholesterol, and Cancer

A player largely omitted in the study of nsPEF’s effect on cells is cholesterol. This molecule is of vital importance to membrane physicochemical properties, but scarce knowledge relates the effect of cholesterol concentration with the formation of nanopores due to the application of nsPEF protocols. Moreover, the abundant evidence described on Section 3 strongly supports the inclusion of cholesterol in future studies for a better comprehension of its role during the application of nsPEF protocols. This will be, in fact, important knowledge considering different cell types exhibit a variety of cholesterol concentrations in their membranes. In particular, further studies should pay attention to understanding how cholesterol changes important physicochemical properties of membranes. When a pore membrane is formed, phospholipids migrate to the pore center in order to hide their hydrophobic carbon chains, exposing their polar groups to the solvent in order to equilibrate the forming pore [270]. Despite being a spontaneous process, the reorganization of phospholipids has an energetic cost associated with the breaking of van der Waals forces between carbon aliphatic chains participating in this rearrangement. This energy penalty per unit length of pore circumference is known as edge tension and denotes a driving force tending to close transient pores [271]. Edge tension is closely related to two important parameters guiding nanopore formation: (i) the membrane charge necessary to induce the nanopore, which is related to nsPEF intensity and duration; and (ii) the lifetime of nsPEF nanopores, which is related to the auto-healing capacity of lipid bilayer structures. Therefore, measuring edge tension is important not only for nsPEF research, but also to better understand various biological events and physicochemical processes occurring in membranes. Of note, MD simulation studies have being used to propose edge tension values [272,273].

As seen in the previous paragraph, the concentration of cholesterol has a strong impact on the edge tension of cellular membranes. Therefore, the presence of cholesterol should also have important consequences for the modulation of nsPEF’s effects in cells. Consistently, cholesterol content together with the phospholipid profile have both been proposed as important factors to explain nsPEF selectivity for different cell types [274]. Of note, available literature suggests that various solid tumors and malignancies present a dysregulated cholesterol metabolism, a characteristic that may be related to the high sensitivity of these cells to the application of nsPEF protocols [275]. Thus, the phospholipid profile in different cancer cell lines is notoriously altered when compared with their non-cancerous counterparts—an important prognosis of cancer malignancy [276,277,278,279,280]. Interestingly enough, the presence of lipid rafts, i.e., membrane domains rich in cholesterol, is also scarcely explored with regard to its relationship to nsPEF’s effects. Abundant literature suggests that the presence of lipid rafts is crucial to anchor ion channels and other transmembrane proteins [281,282]. On top of that, the increased amount of phospholipids in cancer cells occurs mainly on regions forming lipid rafts [280,283]. Even more, the dysregulation of lipid rafts occurring in cancer promotes cell transformation, tumor progression, and metastasis [280]. Considering the available evidence, dissecting the role of cholesterol and phospholipid profiles during the application of nsPEF protocols could be crucial to better understand the sensitivity of cancer cells to this technology—a necessary step towards the development of novel nsPEF-based cancer therapies.

## 9. Conclusions

Despite the controversy in the academic community arising from the timescale in which nsPEF effects are elicited, the key effect at the cellular level is, undoubtedly, the change in Ca${}^{2+}$ homeostasis. Whether this change is due to the formation of membrane nanopores either on the plasma membrane or internal membranes is still a matter of debate and probably dependent on the parameters of the applied protocols. On top of that, abundant evidence supports the notion that the formation of membrane nanopores is linked to the activation of VG channels. Moreover, recently published data coming from MD simulations show that the application of nsPEF-like protocols may also form transient pores within the structure of VGC channels. As a whole, both cell membranes and ion channels should be considered as equally relevant contributors to explain the effects of the application of nsPEF protocols.

In spite of the impressive and massive advancements supporting the development of nsPEF technology, a larger body of research is still needed to better understand the fundamental biophysical principles governing the effects of nsPEF. A better understanding of this interesting phenomenon will eventually allow its translation into a broader and more robust set of applications. To this end, both public and private parties have to become aware of the exceptional capabilities of nsPEF technology and its suitability to be used in both industry and human health.

## Figures and Tables

**Figure 1 ijms-23-06158-f001:**
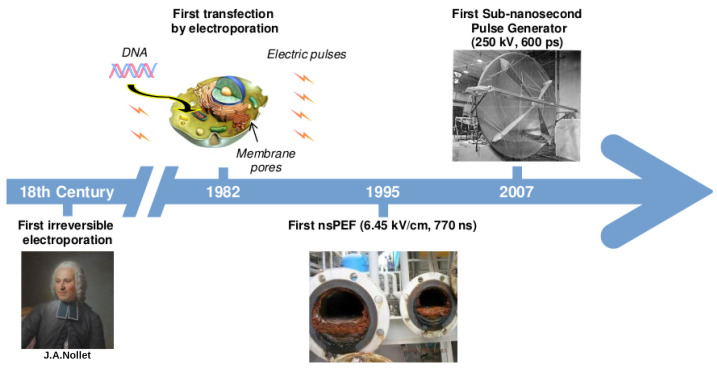
Timeline of main events in the development of electric pulse technology. The first application of electric pulses was recorded in 1754 with the experiments performed by J. A. Nollet. Two centuries later, in 1982, E. Neumann et al. [2] coined the term electroporation to describe the use of electric pulses to create membrane pores allowing the insertion of genetic material into cells. Afterwards, in 1995, Schoenbach et al. [12] developed the first nsPEF technology to prevent biofouling of cooling systems. Lately, the construction of an IRA in 2007 by Heeren et al. [13], allowed the application of sub-nanosecond pulses.

**Figure 2 ijms-23-06158-f002:**
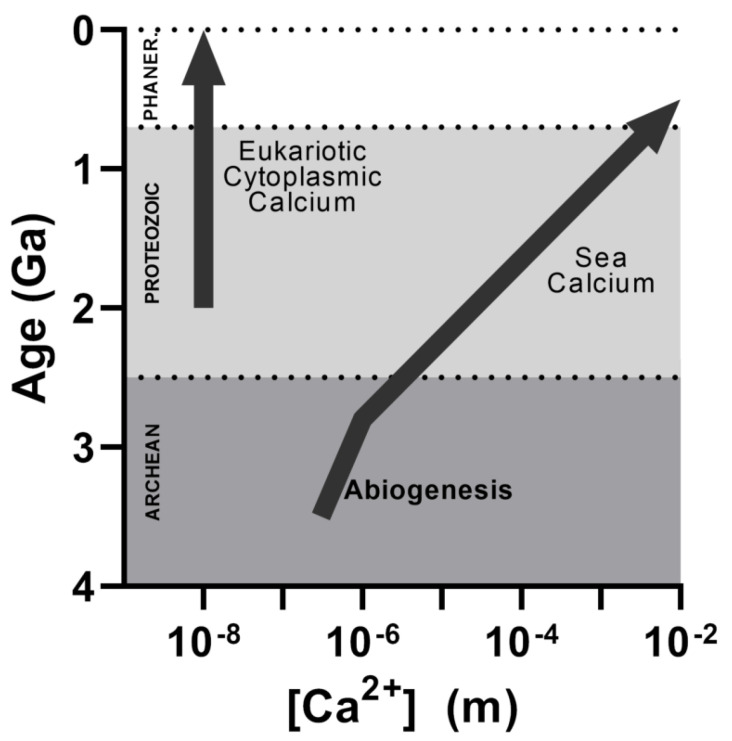
Calcium concentration as a function of time in the sea and in cytoplasmic eukaryotic cells.

**Figure 3 ijms-23-06158-f003:**
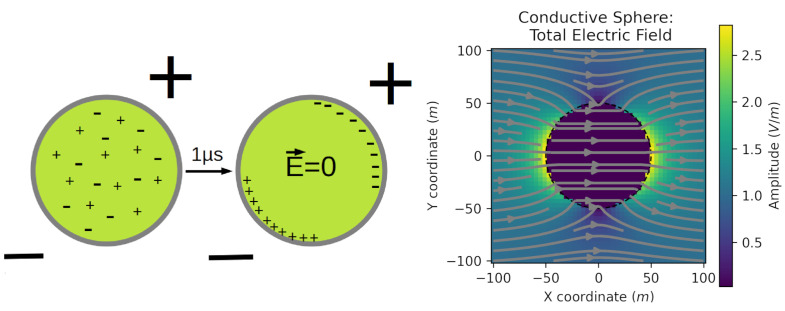
Schematic representation of the movement of charges inside a simplified model of a cell containing positive and negative charges under the application of an external electric field. After a suitable elapsed time, for instance 1 μs, the movement of charges reaches an equilibrium, resulting in the electric field inside the cell being nullified. Right panel showing the total electric field extracted from https://em.geosci.xyz (accessed on 27 April 2022).

**Figure 5 ijms-23-06158-f005:**
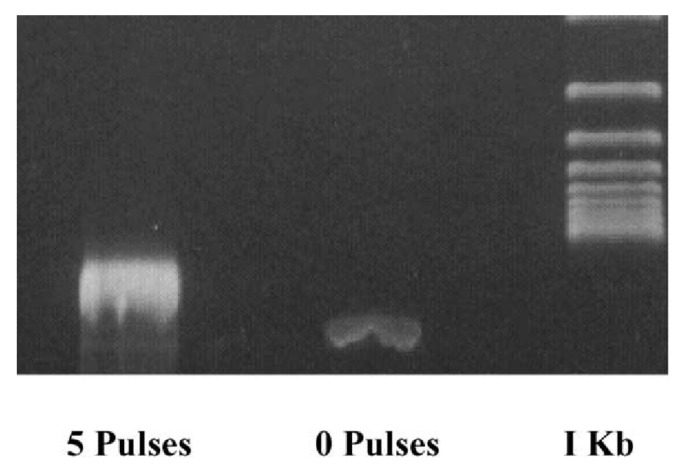
Electrophoresis of DNA extracted from Jurkat cells right after nsPEF (60 kV/cm, 60 ns, 5 pulses). The appearance of a smeared DNA band in the first lane is congruent with DNA damage induced by nsPEF. Taken from [247]. Reproduced with permission.

**Figure 6 ijms-23-06158-f006:**
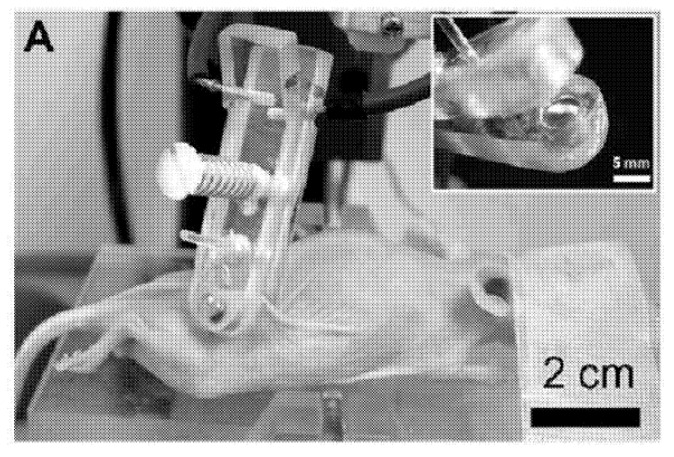
nsPEF device applied on SKH-1 hairless mouse to abolish melanoma cancer. Figure extracted from the patent titled “Nanosecond pulsed electric fields cause melanomas to self-destruct”. ID US20180200510A1.

**Figure 7 ijms-23-06158-f007:**
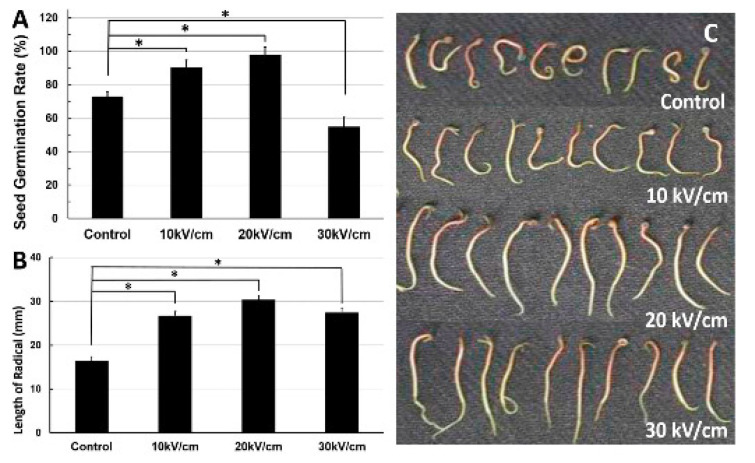
Effects of different intensities of nsPEF on the seed germination rate and radical length of *Haloxylon ammodendron*. (**A**): Seed germination rate at different electric fields. (**B**): Length of radical at different electric fields. (**C**): Image of radical length at different electric fields. Taken from the journal *Plasma Processes and Polymers*, article “Early Growth Effects of Nanosecond Pulsed Electric Field (nsPEFs) Exposure on *Haloxylon ammodendron*”. Copyright Wiley–VCH GmbH. Reproduced with permission.

**Table 1 ijms-23-06158-t001:** Examples of both theoretical and experimental studies exploring nanopore formation upon the application of nsPEF.

nsPEF (kV/cm)	Pulse Time (ns)	Cell Line Used	Observed Effect	Year and Citation
13.5	50	HL-60 leukemia cells	nsPEF affects the nucleus but not the plasma membrane	1997 [47]
60	60	Theoretical cell model including representations of several organelles	nsPEF goes through cell membrane, extensively penetrating organelles	2006 [48]
∼150	10–100	Sp2, mouse murine myeloma cells	The cytoplasmic membrane is capable of withstanding nsPEF application, suggesting that permeabilization of organelles is the main effect	2001 [49]
53	60	Human neutrophil and eosinophil cells	nsPEF induces poration of eosinophils’ intracellular granules. This occurs without permanent disruption of the cytoplasmic membrane. Of note, neutrophils show no changes	2001 [50]
4–15	60	HL-60 leukemia cells	nsPEF induces a rise in cytoplasmic Ca${}^{2+}$ concentration without incorporation of external propidium iodide and using a Ca${}^{2+}$-free media, suggesting no cell membrane poration, but poration of internal membranes	2004 [51]
20–80	4	Chromaffin cells	nsPEF induces a rise in cytoplasmic Ca${}^{2+}$ concentration not affected by the depletion of intracellular calcium storage with either caffeine or thapsigargin, being completely prevented by the presence of EGTA (a Ca${}^{2+}$ chelator) in the extracellular medium	2008 [52]
22–24	60	GH3 murine pituitary, PC-12 murine adrenal, and Jurkat cells (immortalized human T-lymphocytes)	nsPEF induces a long-lasting effect (∼100 s) on cytoplasmic membrane permeabilization that can be monitored by patch-clamp	2007 [53]
2.4–4.8	600	GH3 and CHO-K1 Chinese hamster ovary cells	nsPEF induces an incremental increase in cell conductance, attributed to the formation of ion-channel-like nanopores in the cytoplasmic membrane with a maximum width of 1 nm in both studied cell lines. The size was proposed because the membrane remained mostly impermeable to propidium iodine	2009 [3]

**Table 2 ijms-23-06158-t002:** Examples of studies demonstrating effects of nsPEF on VGCs.

nsPEF (kV/cm)	Pulse Time (ns)	Cell Line or Tissue Used	Observed Effect of nsPEF	Year and Citation
3.1	150–400	Bovine chromaffin cells	Similar results to [186]. Bagalkot et al. in 2019 incorporated a symmetrical bipolar pulse (a second identical pulse but with opposite polarity) that attenuated Ca${}^{2+}$ entry across possible nanopores while preserving Ca${}^{2+}$ influx through VGCCs [189].	2018 [189,190]
190	0.5	GH3, CHO-K1, and NG108 cells (murine neuroblastoma–rat glioma hybrid)	This sub-nanosecond electric pulse activated VGCCs on GH3 and NG108 cells (which express multiple types of VGCCs) and CHO-K1 cells (no VGCC expression). Trains of up to 100 pulses did not change the cytoplasmic Ca${}^{2+}$ concentration (followed by Fura-2 imaging) in CHO-K1 cells, while in GH3 and NG108, a single pulse significantly increased it. Trains of 100 pulses increased cytoplasmic Ca${}^{2+}$ concentration to 379 ± 33 nM in GH3 and 719 ± 315 nM in NG108. To corroborate that Ca${}^{2+}$ is passing through a VGCC and not nanopores, they used verapamil (L-type VGCC blocker) and ω–conotoxin (wide-spectrum N, P, and Q type VGCC blocker). They observed 80-100% inhibition of Ca${}^{2+}$ uptake with both VGCC blockers.	2015 [34]
2.3	300	HEK293 cells	In cells with and without assembled Ca${}_{v}$1.3 L-type VGCC, the nsPEF pulse caused a lasting (>80 s) increase in membrane conductance for all cells. Although the elicited membrane potential did not depolarize enough for VGCC activation, the increase in conductance in cells that expressed VGCC was about two-fold greater than in cells which did not. This result suggests an important role of VGCC in the increase in cytoplasmic Ca${}^{2+}$ concentration induced by nsPEF.	2018 [191]
1.6–1.9	200	E18 rat hippocampal neurons	Using fast optical membrane potential imaging, it was shown that a single nsPEF pulse was able to trigger a single action potential 4–6 ms after the nsPEF pulse in 40% of neurons. The addition of tetrodotoxin (selective sodium channel blocker) to cell media abolished the induced nsPEF action potential, demonstrating that nsPEF managed to activate VGNCs.	2017 [188]
3.3–8.8	12	*Xenopus laevis* peripheral nerve	Using thousands of nsPEF pulses, nerve excitation was achieved without electroporation for the first time. The nerves did not register cumulative damage, as refractory properties were not affected. The authors claimed that their data proved that VGNC are activated by nsPEFs and also manifested that nsPEFs are a promising tool for biomedical applications.	2010 [186]

## Data Availability

Not applicable.

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
