# Peer review of "Nanosecond Pulsed Electric Field (nsPEF): Opening the Biotechnological Pandora’s Box"

_ijms, 2022, doi:10.3390/ijms23116158_

Round 1

Reviewer 1 Report

This review paper was well-organized and contained many important information which maybe interested in many readers in same research field. Therefore, I strongly recommend the paper for publication after some minor editorial (ex. page 3, left column, line 3 from the bottom; The alphabet "q" should be changed to italic font, because the "q" means the charge which is physical quantity).

Reviewer 2 Report

Review on manuscript entitled “Nanosecond Pulsed Electric Field (nsPEF): opening the biotechnological Pandora’s box”.

As a review work the manuscript gives a really interesting and useful logical flow in the pulsed electric field stimulation. Authors have created an informative work, with clear structure, motivated by the latest discoveries in the field and inspirational future trends. So I found it acceptable for publication and interesting for the readers of IJMS.

Here are some critical remarks:

Modeling and simulation is the actual wide-open door in bio-electric stimulations. Unfortunately, it is somehow neglected in the manuscript. This is very strange, when reading in the Acknowlegments section National Laboratory for High Performance Computing (NLHPC). In my opinion view on modeling must be enlarged.

Around equation (1) is necessary to explain that this is for tissue polarization dynamics modeling, now it is not clear what are these capacitances and resistances. Tau must be referred as time-constant always, now it has been mentioned just as constant which is misleading.

Chapter “5.2 The time-scale controversy behind ion channel activation by nsPEF protocols: the role of MD simulations” is really interesting, but does not dive deep in the MD modeling. Time scale is recognized as a main complexity barrier, emerging from fs scale of MD and ms scale of electric models. Please note that electric force is applied only at ions and charged particles, but not on atoms at is mentioned several times.

Discussion chapter will be welcomed, summarizing major directions and trends.

Conclusion must be enlarged.

Minor corrections needed:

“Where” after an equation is without indentation.

Page 11, line 31, left- XX V/nm during XX ns (COMPLETAR) is unfinished.

Figure 6 is referred to US20180200510A1, actually it is strange to have a color picture in a patent. Probably source is different.
